# HOW MANY WEIGHTS ARE ENOUGH : CAN TENSOR FACTORIZATION LEARN EFFICIENT POLICIES ?

## ABSTRACT

Deep reinforcement learning requires a heavy price in terms of sample efficiency and overparameterization in the neural networks used for function approximation. In this work, we employ *tensor factorization* in order to learn more compact representations for reinforcement learning policies. We show empirically that in the low-data regime, it is possible to learn online policies with 2 to 10 times less total coefficients, with little to no loss of performance. We also leverage progress in second order optimization, and use the theory of *wavelet scattering* to further reduce the number of learned coefficients, by foregoing learning the topmost convolutional layer filters altogether. We evaluate our results on the Atari suite against recent baseline algorithms that represent the state-of-the-art in data efficiency, and get comparable results with an order of magnitude gain in weight parsimony.

## 1 INTRODUCTION

The successes of reinforcement learning (thereafter 'RL'), and specifically deep RL, come at a heavy computational price. It is well known that achieving human-level performance in domains such as Atari (Sutton & Barto, 2018; Mnih et al., 2013; Hessel et al., 2017) requires hundreds of millions of frames of environment interaction. As such, the problem of sample efficiency in RL is of critical importance. Several tracks of concurrent research are being investigated, and have reduced by orders of magnitude the number of environment interactions required for good performance beyond the previous benchmark of biologically-inspired episodic control methods (Blundell et al., 2016; Pritzel et al., 2017) to a couple hours of human gameplay time (van Hasselt et al., 2019; Kaiser et al., 2019).

However, while the data-efficiency of RL methods has seen recent drastic performance gains, the function approximators they use still require millions of learned weights, potentially still leaving them heavily overparameterized. Independently motivated by biological facts like the behavioural readiness of newborn animals, several authors (Gaier & Ha, 2019; Cuccu et al., 2018; Wang et al., 2019) have recently looked at doing away with learning so many weights for RL tasks. Smaller networks not only train faster, but may yet offer another avenue for gains in the form of better generalization (Zhang et al., 2016). Recent work from Gaier & Ha (2019) studies the effect of inductive bias of neural architectures in RL ; they forego training altogether, but transfer networks that only obtain 'better than chance performance on MNIST'. In similar fashion, Wang et al. (2019) investigate the effect of random projections in the restricted setting of imitation learning. Finally, Cuccu et al. (2018) manage human-level performance on the Atari suite using a separate dictionary learning procedure for their features, bypassing the usual end-to-end learning paradigm. The perspective of neural architecture search applied to RL appears difficult, if not computationally inextricable.

Concurrently, the study of biologically-inspired models of learning has exhibited two mathematical characterizations that might be critical in explaining how biological learning takes place so efficiently. First, the low-rank properties of learned perceptual manifolds (Chung et al., 2018; 2016) are giving rise to a rich theory borrowing from statistical physics. Second, another well known line of work has identified Gabor filters, and more generally wavelet filter-like structures, in the actual visual cortex of animals (Jones & Palmer, 1987), and linked those to sparsity-promoting methods and dictionary learning (Olshausen & Field, 1996; 1997; Hyvärinen & Hoyer, 2001). But these breakthroughs have not, so far, been reflected as inductive priors in the shape of modifications in deep RL neural networks architectures, which remain fairly fixed on the Atari domain.

Therefore the following questions remain: how parsimonious do function approximators in RL need to be, in order to maintain good performance? And can we be at once *sample-efficient* and *weight-efficient* ? In this work, we turn to the mathematical theories of tensor factorization (Cichocki et al., 2009), second-order optimization (Amari, 1998; Martens & Grosse, 2015) and wavelet scattering (Mallat, 2011) to answer this question positively and empirically, in a model-free setting.

We propose to use these methods in order to save weights and therefore favour convergence of policies:

- We replace dense, fully-connected layers with *tensor regression layers*.
- Optionally, we replace the topmost layer in the convolutional architecture with a *scattering* layer; the deeper convolutional layers are left untouched.
- The (positive) impact of second-order optimization is also evaluated.

To the best of our knowledge, this is the first time those fields have been combined together in this context, and that tensor factorization is applied to deep RL.

## 2 BACKGROUND & RELATED WORK

### 2.1 DEEP REINFORCEMENT LEARNING

We consider the standard Markov Decision Process framework as in Sutton & Barto (2018). This setting is characterised by a tuple $\langle S, A, T, R, \gamma \rangle$, where $S$ is a set of states, $A$ a set of actions, $R$ a reward function that is the immediate, intrinsic desirability of a certain state, $T$ a transition dynamics and $\gamma \in [0, 1]$ a discount factor. The purpose of the RL problem is to to find a policy $\pi$, which represents a mapping from states to a probability distribution over actions, that is optimal, i.e., that maximizes the expected cumulative discounted return $\sum_{k=0}^{\infty} \gamma^k R_{t+k+1}$ at each state $s_t \in S$. In Q-learning, the policy is given implicitly by acting greedily or $\epsilon$-greedily with respect to learned *action-value functions* $q^\pi(s, a)$, that are learned following the Bellman equation. In *deep Q-learning*, $q_\theta$ becomes parameterized by the weights $\theta$ of a neural network and one minimizes the expected Bellman loss :

$$\mathbb{E} \left( R_{t+1} + \gamma_{t+1} \max_{a'} q_\theta \left( S_{t+1}, a' \right) - q_\theta \left( S_t, A_t \right) \right)^2$$

In practice, this is implemented stochastically via uniform sampling of transitions in an experience replay buffer, as is done in the seminal paper Mnih et al. (2013). Several algorithmic refinements to that approach exist. First, Double Q-learning (van Hasselt et al., 2015) proposes to decouple learning between two networks in order to alleviate the Q-value overestimation problem. Second, *dueling* Q-networks (Wang et al., 2015) explicitly decompose the learning of an action-value function $q_\theta(s, a)$ as the sum of an action-independent state-value, much like what is traditionally done in policy gradient methods (Sutton & Barto, 2018), implemented via a two-headed neural network architecture. Finally, *prioritized* RL (Schaul et al., 2015) proposes to replace the uniform sampling of transitions in the experience replay buffer with importance sampling, by prioritizing those transitions that present the most Bellman error (those transitions that are deemed the most 'surprising' by the agent). Fortunato et al. (2017) use extra weights to learn the variance of the exploration noise in a granular fashion, while Bellemare et al. (2017) propose to learn a full *distribution* of action-values for each action and state. Combined, those methods form the basis of the Rainbow algorithm in Hessel et al. (2017).

### 2.2 TENSOR FACTORIZATION

Here we introduce notations and concepts from the tensor factorization literature. An intuition is that the two main decompositions below, *CP* and *Tucker* decompositions, can be understood as multilinear algebra analogues of SVD or eigendecomposition.

**CP decomposition.** A tensor $\mathcal{X} \in \mathbb{R}^{I_1 \times I_2 \times \cdots \times I_N}$, can be decomposed into a sum of $R$ rank-1 tensors, known as the Canonical-Polyadic decomposition, where $R$ is the rank of the decomposition. Its purpose is to find vectors $\mathbf{u}_k^{(1)}, \mathbf{u}_k^{(2)}, \cdots, \mathbf{u}_k^{(N)}$, for $k = [1 \ldots R]$, as well as a vector of weights

$\boldsymbol{\lambda} \in \mathbb{R}^R$ such that:

$$\mathcal{X} = \sum_{k=1}^{R} \lambda_k \underbrace{\mathbf{u}_k^{(1)} \circ \mathbf{u}_k^{(2)} \circ \cdots \circ \mathbf{u}_k^{(N)}}_{\text{rank-1 components}}$$

**Tucker decomposition.** A tensor $\mathcal{X} \in \mathbb{R}^{I_1 \times I_2 \times \cdots \times I_N}$, can be decomposed into a low rank approximation including a core $\mathcal{G} \in \mathbb{R}^{R_1 \times R_2 \times \cdots \times R_N}$ and a set of projection factors $\left(\mathbf{U}^{(0)}, \cdots, \mathbf{U}^{(N-1)}\right)$, with $\mathbf{U}^{(k)} \in \mathbb{R}^{R_k, \hat{I}_k}, k \in (0, \cdots, N-1)$ that, when projected along the corresponding dimension of the core, reconstruct the full tensor $\mathcal{X}$. The tensor in its decomposed form can be written:

$$\mathcal{X} = \mathcal{G} \times_1 \mathbf{U}^{(1)} \times_2 \mathbf{U}^{(2)} \times \cdots \times_N \mathbf{U}^{(N)} = \left[\mathcal{G}; \mathbf{U}^{(1)}, \cdots, \mathbf{U}^{(N)}\right]$$

**Tensor regression layer.** For two tensors $\mathcal{X} \in \mathbb{R}^{K_1 \times \cdots \times K_x \times I_1 \times \cdots \times I_N}$ and $\mathcal{Y} \in \mathbb{R}^{I_1 \times \cdots \times I_N \times L_1 \times \cdots \times L_y}$, we denote by $\langle \mathcal{X}, \mathcal{Y} \rangle_N \in \mathbb{R}^{K_1 \times \cdots \times K_x \times L_1 \times \cdots \times L_y}$ the contraction of $\mathcal{X}$ by $\mathcal{Y}$ along their $N$ last modes; their generalized inner product is

$$\langle \mathcal{X}, \mathcal{Y} \rangle_N = \sum_{i_1=1}^{I_1} \sum_{i_2=1}^{I_2} \cdots \sum_{i_n=1}^{I_N} \mathcal{X}_{\ldots,i_1,i_2,\ldots,i_n} \mathcal{Y}_{i_1,i_2,\ldots,i_n,\ldots}$$

This makes it possible to define a *tensor regression layer* (Kossaifi et al., 2017b) that is differentiable and learnable end-to-end by gradient descent. Let us denote by $\mathcal{X} \in \mathbb{R}^{I_1 \times I_2 \times \cdots \times I_N}$ the input activation tensor for a sample and $\mathbf{y} \in \mathbb{R}^{I_N}$ the label vector. A tensor regression layer estimates the regression weight tensor $\mathcal{W} \in \mathbb{R}^{I_1 \times I_2 \times \cdots \times I_N}$ under a low-rank decomposition. In the case of a Tucker decomposition (as per our experiments) with ranks $(R_1, \cdots, R_N)$, we have :

$$\mathbf{y} = \langle \mathcal{X}, \mathcal{W} \rangle_N + \mathbf{b} \qquad \text{with } \mathcal{W} = \mathcal{G} \times_1 \mathbf{U}^{(1)} \times_2 \mathbf{U}^{(2)} \cdots \times_N \mathbf{U}^{(N)}$$

as $\mathcal{G} \in \mathbb{R}^{R_1 \times \cdots \times R_N}$, $\mathbf{U}^{(k)} \in \mathbb{R}^{I_k \times R_k}$ for each $k$ in $[1 \ldots N]$ and $\mathbf{U}^{(N)} \in \mathbb{R}^{1 \times R_N}$.

## 2.3 Wavelet scattering

The *wavelet scattering transform* was originally introduced by Mallat (2011) and Bruna & Mallat (2012) as a non-linear extension to the classical wavelet filter bank decomposition. Its principle is as follows. Denoting by $x \circledast y[n]$ the 2-dimensional, circular convolution of two signals $x[n]$ and $y[n]$, let us assume that we have pre-defined two wavelet filter banks available $\left\{\psi_{\lambda_1}^{(1)}[n]\right\}_{\lambda_1 \in \Lambda_1}$ $\left\{\psi_{\lambda_2}^{(2)}[n]\right\}_{\lambda_2 \in \Lambda_2}$, with $\lambda_1$ and $\lambda_2$ two frequency indices. These wavelet filters correspond to high frequencies, so we also give ourselves the data of a lowpass filter $\phi_J[n]$. Finally, and by opposition to traditional linear wavelet transforms, we also assume a given nonlinearity $\rho(t)$. Then the scattering transform is given by coefficients of order 0,1, and 2, respectively :

$$S_0 x[n] = x \circledast \phi_J[n]$$

$$S_1 x[n, \lambda_1] = \rho\left(x \circledast \psi_{\lambda_1}^{(1)}\right) \circledast \phi_J[n] \quad \lambda_1 \in \Lambda_1$$

$$S_2 x[n, \lambda_1, \lambda_2] = \rho\left(\rho\left(x \circledast \psi_{\lambda_1}^{(1)}\right) \circledast \psi_{\lambda_2}^{(2)}\right) \circledast \phi_J[n] \quad \lambda_1 \in \Lambda_1, \lambda_2 \in \Lambda_2\left(\lambda_1\right)$$

This can effectively be understood and implemented as a two-layer convolutional neural network whose weights are not learned but rather frozen and given by the coefficients of wavelets $\psi$ and $\phi$ (with Gabor filters as a special case (Mallat, 1998)). The difference with traditional filter banks comes from the iterated modulus/nonlinear activation function applied at each stage, much like in traditional deep learning convolutional neural networks. The generic mathematical definition involves order $n$ iterated scatterings, in the vein of $S_i x$ above, but sometimes restricts nonlinearity $\rho$ to be a modulus function $|\cdot|$. In practice, the potential of scattering transforms to accelerate learning by providing ready-made convolutional layers has been investigated in Oyallon et al. (2013) and Oyallon et al. (2018) and is a subject of active ongoing research. Scattering will be the second of our weight-saving methods.

## 2.4 SECOND ORDER OPTIMIZATION WITH K-FAC

While stochastic gradient descent is usually performed purely from gradient observations derived from auto-differentiation, faster, second order optimization methods first multiply the weights' $\theta$ gradient vector $\nabla_\theta$ by a preconditioning matrix, yielding the weight update $\theta_{n+1} \leftarrow \theta_n - \eta G_n^{-1} \nabla_\theta$, with $\eta$ a step size. In the case of second order methods, the matrix $G_n^{-1}$ is chosen to act as a tractable iterative approximation to the inverse Hessian or Empirical Fisher Information Matrix (Amari, 1998) of the neural network model in question. Kronecker-factored approximate curvature or K-FAC (Martens & Grosse, 2015) enforces a Kronecker decomposition of the type $G = A \otimes B$, with $A$ and $B$ being smaller, architecture-dependent matrices. Unlike the above methods, K-FAC *has* been applied as a plug-in in the deep RL literature and shown to promote both anytime convergence properties as well as terminal accuracies (Wu et al., 2017).

## 3 OBSERVATIONS AND METHODS

### 3.1 EXPLORING TRAINED AGENTS

**Stability of trained dense layers eigenvalues.** In order to assess experimentally if tensor factorization can make sense in RL, we investigate the eigenvalues of the dense layers of a deep RL agent. Unlike the traditional supervised learning setting, the input data distribution to RL function approximators shifts as the agent explores its environment; as such, concentration properties of the eigenvalues of the linear layers cannot be guaranteed all the way throughout training. Since conditioning techniques such as batch normalization (Ioffe & Szegedy, 2015) are rarely used in deep RL, this is all the more important. Our experiments (see figure 1) show that the distribution of eigenvalues does not seem to widen significantly, at least during the initial phases of training we care about.

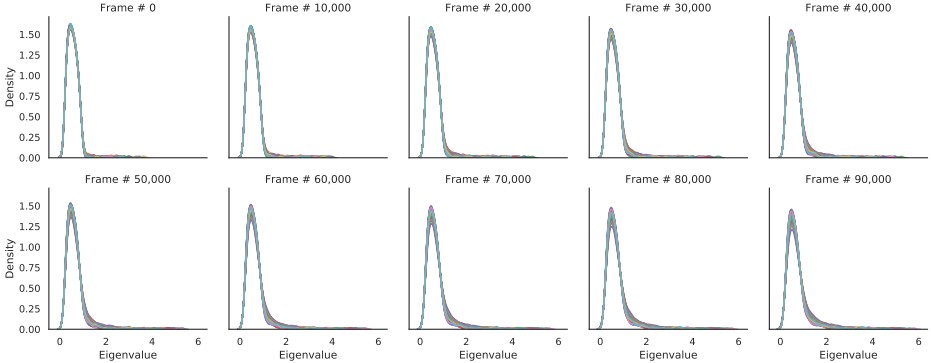

Figure 1: Eigenvalue histograms of the value-based linear layer during training. 50 agent runs data-efficient Rainbow (van Hasselt et al., 2019), of 100,000 steps on the Atari game *Road Runner*.

Furthermore and interestingly, it does not seem to deviate significantly from the one observed at initialization. All together, this suggests there might be some merit in learning low-rank policies.

### 3.2 ARCHITECTURAL MODIFICATIONS

**Baseline.** We then proceed to build upon the well-known Rainbow algorithm (Hessel et al., 2017). Rainbow uses a fairly standard shallow convolutional architecture like the seminal DQN paper of Mnih et al. (2013), and is an oft-cited baseline on the Atari suite. In spite of its performance, Rainbow often requires dozens of millions of a single game's frames in order to perform well. Very recently, a 'data-efficient' efficient version of Rainbow has been proposed by van Hasselt et al. (2019), with a view to match or beat the latest state-of-the-art results achieved by model-based RL methods. This is achieved with no major change in network architecture, but via a selection of mildly hand-tuned hyperparameters favouring efficiency against wall-clock running time (see appendix). We do take this as a baseline.

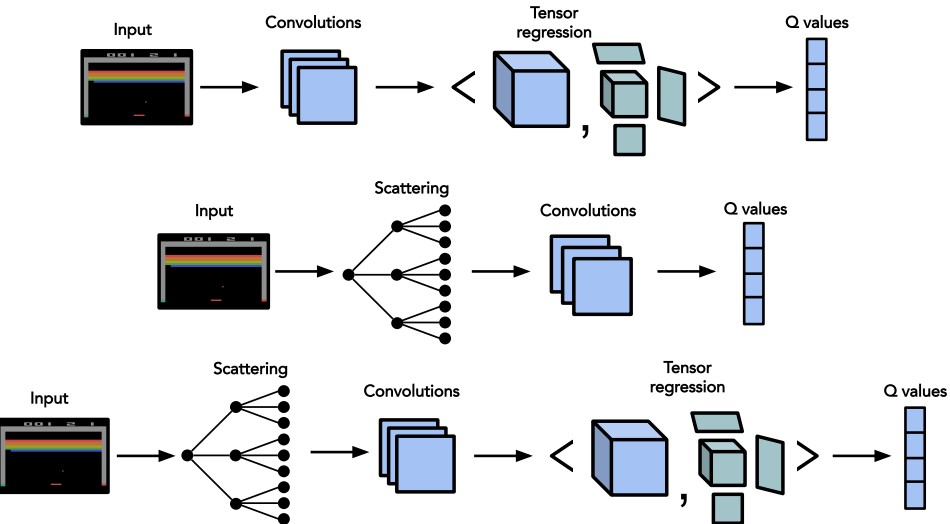

Figure 2: Our architectural approach consists in replacing hidden layers in deep RL agents with tensor regression (top). Optionally we substitute the topmost convolutional layer with scattering (middle), and combine both methods (bottom).

**Changes.** We modify the architecture of the neural network function approximators used, in accordance with the principles described above, combining them to reflect inductive biases promoting fewer learnable parameters:

- We replace the fully-connected, linear layers used in Rainbow and data-efficient Rainbow with tensor regression layers (Kossaifi et al., 2017b) in order to learn *low-rank* policies (ranks in appendix).

- We use either the K-FAC second order stochastic optimizer, or the standard ADAM optimizer (Kingma & Ba, 2014). Optimization with K-FAC yields better results *ceteris paribus* and therefore works to counter performance loss due to using fewer weights.

- We combine the two methods with various rank (and therefore weight compression) ratios, targetting sensible compression ratios guided by deep learning intuition; and evaluate those on the same subset of Atari games as both van Hasselt et al. (2019); Kaiser et al. (2019).

- When possible, we replace the first convolutional layer in the approximating neural network with a *scattering* layer for further gains in terms of learnable weights. In that way, we investigate the impact of not actually learning one of the convolutional layer weights. Deep RL, paradoxically, tends to use shallow network architectures with two or three convolutional layers only, which makes it very fit for scattering methods. But since learned convolutional features discriminate well between high and low rewards, the promise of fully unsupervised scattering layers seems remote, without resorting to further ad-hoc methods like dictionary learning. Therefore we limit ourselves to one-step (one single layer) scatterings.

This is illustrated in figure 2. We then proceed to evaluate the merit of these changes.

## 4 EXPERIMENTAL RESULTS

### 4.1 PRIORITIZED TENSORIZED DQN

**Proof of concept.** We begin with showing proof of concept on the simple *Pong* Atari game. Our experimental setup consists in our own implementation of *prioritized double DQN* as a baseline (Schaul et al., 2015; van Hasselt et al., 2015). We replaced the densely connected layer of the original DQN architecture with a tensor regression layer implementing Tucker decomposition for different Tucker ranks, yielding different network compression factors.

**Qualitative behaviour.** First results, both in terms of learning performance and compression factor, can be seen in figure 3. Our two main findings are that first, the final performance of the agent remains unaffected by the tensor factorization, even with high compression rates - *with respect to all network weights* - of five times. In line with intuition, larger compression rates do however cause more delays in learning. Second, tensor factorization negatively affects stability during training - in tough compression regimes, the plateauing phases of learning curves feature occasional noisy drawdowns, illustrating the increased difficulty of learning, as seen in figure 4. Interestingly, approximation errors incurred by tensor regression noise do sometimes have poor consequences illustrated by those drawdowns, but overall seem to behave as additional exploration noise.

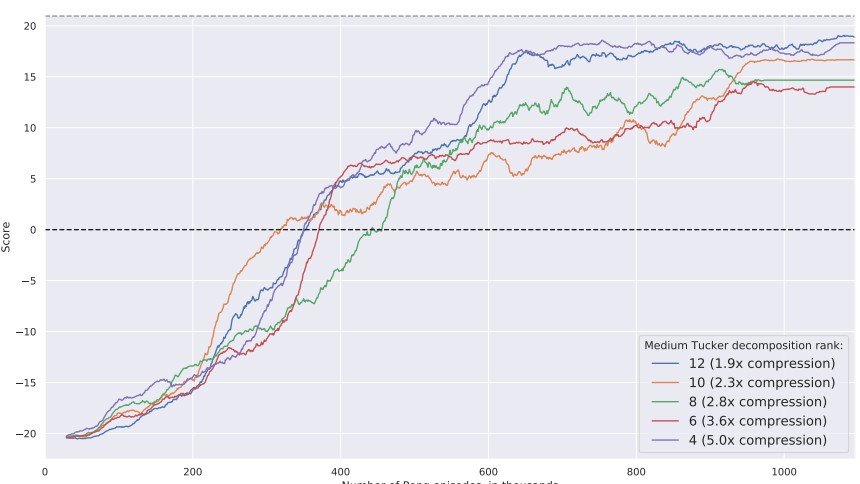

Figure 3: Prioritized tensorized DQN on Atari Pong. Original learning curve versus several learning curves for five different Tucker ranks factorizations and therefore parameter compression rates (3 different random seeds each, with a 30 episodes moving average for legibility). Best viewed in colour.

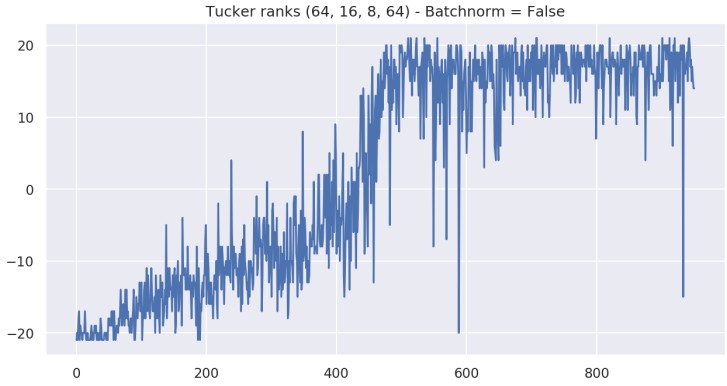

Figure 4: Focus on a typical single run of the tensorized DQN learning (score vs. number of thousand episodes). The overall shape of the typical learning curve is preserved, but drawdowns in the plateauing phase do appear.

## 4.2 DATA-EFFICIENT RAINBOW ON ATARI

**Evaluation protocol.** For all our Atari experiments, we used OpenAI Gym (Brockman et al., 2016), and a combination of PyTorch (Paszke et al., 2017), TensorLy (Kossaifi et al., 2016) and Kymatio (Andreux et al., 2018) for auto-differentiation. We evaluated our agents in the low-data regime of

100,000 steps, on half the games, with 3 different random seeds for reproducibility (Henderson et al., 2017), taking the data-efficient Rainbow agent (van Hasselt et al., 2019) as our baseline. Our specific hyperparameters are described in appendix. We report our results in tables 1 and 2.

| Game | SimPLe | Rainbow | Denoised | TRL 2.5x | TRL 5x | TRL 10x |
|------|--------|---------|----------|----------|--------|---------|
| alien | 405 | **740** | 684 | 688 | 454 | 566 |
| amidar | 88 | **189** | 154 | 118 | 86 | 84 |
| assault | 369 | 431 | 321 | **543** | 521 | 513 |
| asterix | **1090** | 471 | 500 | 459 | 554 | 363 |
| bank_heist | 8 | 51 | 77 | 59 | **134** | 42 |
| battle_zone | 5184 | 10125 | 9378 | **14466** | 13466 | 5744 |
| boxing | **9** | 0.2 | 1 | -2 | -2 | -5 |
| breakout | **13** | 2 | 3 | 2 | 2 | 4 |
| chopper_command | 1247 | 862 | **1293** | 1255 | 1243 | 1106 |
| crazy_climber | **39828** | 16185 | 9977 | 3928 | 4225 | 2340 |
| demon_attack | 170 | **508** | 450 | 362 | 263 | 175 |
| freeway | 20 | **28** | 28 | 26 | 25 | 24 |
| frostbite | 255 | 867 | **1101** | 659 | 912 | 231 |
| gopher | **771** | 349 | 391 | 278 | 255 | 396 |
| hero | 1295 | **6857** | 3013 | 5351 | 3732 | 3321 |
| jamesbond | 125 | **302** | 295 | 215 | 213 | 218 |
| kangaroo | 323 | **779** | 1002 | 804 | 715 | 400 |
| krull | **4540** | 2852 | 2656 | 2333 | 2275 | 2308 |
| kung_fu_master | **17257** | 14346 | 4037 | 9392 | 4764 | 4031 |
| ms_pacman | 763 | **1204** | 1053 | 818 | 838 | 517 |
| pong | **5** | -19 | -20 | -20 | -19 | -21 |
| private_eye | 58 | 98 | 100 | 51 | 100 | **1128** |
| qbert | 560 | **1153** | 672 | 697 | 581 | 733 |
| road_runner | 5169.4 | **9600** | 5426 | 6965 | 3914 | 1319 |
| seaquest | 371 | 354 | **387** | 345 | 350 | 287 |
| up_n_down | 2153 | 2877 | **5123** | 2197 | 2302 | 2179 |
| **Average (vs. Rainbow)** | | **100%** | **118%** | **96%** | **90%** | **71%** |

Table 1: Mean episode returns as reported in baselines SimPLe (Kaiser et al., 2019) and data-efficient Rainbow (van Hasselt et al., 2019), versus our agents, on 26 Atari games. *'Denoised'* is the NoisyNet ablation of Rainbow; *'TRL'* shows the performance of the data-efficient Rainbow with tensor regression layers substituted for linear ones.

Table 1 shows proof of concept of the online learning of low-rank policies, with a loss of final performance varying in proportion to the compression in the low-rank linear layers, very much like in the deep learning literature (Kossaifi et al., 2017a;b). The number of coefficients in the original data-efficient Rainbow is of the order of magnitude of 1M and varies depending on the environment and its action-space size. The corresponding tensor regression layer ranks are in appendix, and chosen to target 400k, 200k and 100k coefficients respectively. While individual game results tend to decrease monotonously with increasing compression, we observe that they are noisy due to the nature of exploration in RL, and average scores reported correspond to the intuition that performance seems to decrease fast after a certain overparameterization threshold is crossed. To take this noisy character into account, we take care to be conservative and report the average of the final three episodes of the learned policy after 80k, 90k and 100k steps, respectively.

**Denoised baseline.** So as to not muddy the discussion and provide fair baselines, we do report on the NoisyNet (Fortunato et al., 2017) ablation of Rainbow ('Denoised' columns), as the NoisyLinear layer doubles up the number of coefficients required and actually performs worse in our experiments. Its principle is that the variance of exploration noise represents a criticial tradeoff for performance (too little and one stalls, too much and one risks catastrophic updates), so it is sensible to treat it as a parameter to learn. However, in order to decouple both factors of overparametrization and exploration in the discussion of deep RL performance, we simply use a fixed exploration schedule. This *denoised exploration* baseline is an ablation we can then compare our tensorized methods to.

**Second-order optimization.** We then proceed to assess the impact of second-order optimization in architecture by substituting ADAM optimization for K-FAC, and introducing scattering, in table 2. In spite of our conservative reporting, the efficiency boost from using a second order scheme more than makes up for low-rank approximation error (109% performance) with five times less coefficients than van Hasselt et al. (2019), and learning with a full order of magnitude less coefficients (98% performance) is made possible by the combination of K-FAC and TRL. The results however do show a sharp drop in average performance when scattering is added. Interestingly enough some games perform relatively very well with that method, simultaneously showing proof of viability and of additional work required.

| Game | KFAC+Denoised | KFAC+TRL5x | KFAC+TRL10x | Scattering |
|------|---------------|------------|-------------|------------|
| alien | **996** | 734 | 643 | 441 |
| amidar | **163** | 101 | 98 | 84 |
| assault | **501** | 491 | 496 | 434 |
| asterix | 537 | **549** | 526 | 502 |
| bank_heist | **100** | 73 | 57 | 29 |
| battle_zone | 8622 | **15178** | 6156 | 4311 |
| boxing | **0** | -4 | -1 | -9 |
| breakout | 3 | **3** | 2 | 2 |
| chopper_command | 692 | 611 | **1302** | 441 |
| crazy_climber | **14242** | 12377 | 3546 | 740 |
| demon_attack | 582 | 434 | 318 | **692** |
| freeway | 26 | 26 | 24 | 19 |
| frostbite | **1760** | 718 | 1483 | 654 |
| gopher | **363** | 341 | 265 | 172 |
| hero | 4188 | **6284** | 4206 | 4127 |
| jamesbond | 263 | **327** | 217 | 48 |
| kangaroo | **2085** | 613 | 588 | 391 |
| krull | 2855 | **3441** | 3392 | 772 |
| kung_fu_master | 8481 | **10738** | 7357 | 233 |
| ms_pacman | **1137** | 920 | 867 | 613 |
| pong | -19.3 | **-19** | -19 | -20 |
| private_eye | 56 | **100** | 100 | 0 |
| qbert | **731** | 520 | 538 | 475 |
| road_runner | 4516 | **8493** | 7224 | 1278 |
| seaquest | 349 | 317 | **520** | 213 |
| up_n_down | **2557** | 2291 | 2108 | 993 |
| **Average (vs. Rainbow)** | **114%** | **109%** | **98%** | **55%** |

Table 2: Mean episode returns of our low-rank agents with second-order optimization and scattering. The Scattering column also includes KFAC optimization and TRL 5x, resulting in around 10x total weights efficiency gains.

## 5 CONCLUSION

We have demonstrated that in the low-data regime, it is possible to leverage biologically plausible characterizations of experience data (namely low-rank properties and wavelet scattering separability) to exhibit architectures that learn policies with *an order of magnitude less weights than current state-of-the-art baselines*, essentially without loss of performance, and in an online fashion. In particular, this provides a compelling alternative to methods like policy distillation (Rusu et al., 2015; Czarnecki et al., 2019). We do hope that this will lead to even further progress towards sample efficiency and speedy exploration methods. Further work will, first, focus on thorough evaluation and research of scattering architectures in order to achieve further gains, and second investigate additional, orthogonal biologically-friendly research directions such as promoting sparsity via, for instance, $L^1$ regularization. Finally, we are excited by the potential of tensor factorization to offer shared core tensors for policies in multi-task and meta-learning.

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

# A APPENDIX

## HYPERPARAMETERS AND REPRODUCIBILITY

Hyperparameters are as follows. First, our specific architecture-modified hyperparameters:

| Specific architecture hyperparameters | Value |
| --- | --- |
| Scattering maximum log-scale $J$ | 3 |
| Scattering volume width $M$ | 1 |
| Scattering tensor input shape | (1,4,84,84) |
| Scattering tensor output shape | (1,16,11,11) |
| Scattering type | Harmonic 3D, see Andreux et al. (2018); Eickenberg et al. (2018) |
| Hidden linear layer rank constraint, 2.5x compression | 128 |
| Final linear layer rank constraint, 2.5x compression | 48 |
| Hidden linear layer rank constraint, 5x compression | 32 |
| Final linear layer rank constraint, 5x compression | 48 |
| Hidden linear layer rank constraint, 10x compression | 16 |
| Final linear layer rank constraint, 10x compression | 10 |
| KFAC Tikhonov regularization parameter | 0.1 |
| KFAC Update frequency for inverses | 100 |

Table 3: Our additional, architecture-specific hyperparameters.

Furthermore, we mirror the Data-Efficient Rainbow van Hasselt et al. (2019) baseline:

| Data-efficient Rainbow hyperparameters | Value |
| --- | --- |
| Grey-scaling | True |
| Observation down-sampling | (84, 84) |
| Frames stacked | 4 |
| Action repetitions | 4 |
| Reward clipping | [-1, 1] |
| Terminal on loss of life | True |
| Max frames per episode | 108K |
| Update | Distributional Double Q |
| Target network update period* | every 2000 updates |
| Support of Q-distribution | 51 bins |
| Discount factor | 0.99 |
| Minibatch size | 32 |
| Optimizer | Adam |
| Optimizer: first moment decay | 0.9 |
| Optimizer: second moment decay | 0.999 |
| Optimizer: $\epsilon$ | 0.00015 |
| Max gradient norm | 10 |
| Priority exponent | 0.5 |
| Priority correction** | $0.4 \rightarrow 1$ |
| Hardware | NVidia 1080Ti GPU |
| Noisy nets parameter | 0.1 |
| Training frames | 400,000 |
| Min replay size for sampling | 1600 |
| Memory size | unbounded |
| Replay period every | 1 steps |
| Multi-step return length | 20 |
| Q network: channels | 32, 64 |
| Q network: filter size | 5 5, 5 5 |
| Q network: stride | 5, 5 |
| Q network: hidden units | 256 |
| Optimizer: learning rate | 0.0001 |

Table 4: Data-efficient Rainbow agent hyperparameters, as per van Hasselt et al. (2019).

Our codebase is available on request.

STANDARD DEVIATIONS FOR SCORE RUNS

| Game | Denoised | TRL 2.5x | TRL 10x |
|---|---|---|---|
| alien | 684 ± 7 | 688 ± 123 | 566± 38 |
| amidar | 154 ± 21 | 118 ± 12 | 84± 15 |
| assault | 321 ± 224 | 543 ± 94 | 513± 64 |
| asterix | 500 ± 124 | 459 ± 91 | 363 ± 66 |
| bank_heist | 77 ± 23 | 59 ± 22 | 42 ± 2 |
| battle_zone | 9378 ± 2042 | 14466 ± 2845 | 5744 ± 575 |
| boxing | 1 ± 2 | -2 ± 1 | -5 ± 1 |
| breakout | 3 ± 1.5 | 2 ± 1 | 4 ± 0.3 |
| chopper_command | 1293 ± 445 | 1255 ± 215 | 1106 ± 124 |
| crazy_climber | 9977 ± 3744 | 3928 ± 221 | 2340 ± 595 |
| demon_attack | 450 ± 49 | 362 ± 147 | 175 ± 7 |
| freeway | 28 ± 0.6 | 26 ± 0 | 24 ± 0.5 |
| frostbite | 1101 ± 355 | 659 ± 523 | 231 ± 1 |
| gopher | 391 ± 46 | 278 ± 39 | 396 ± 24 |
| hero | 3013 ± 90 | 5351 ± 1948 | 3321 ± 598 |
| jamesbond | 295 ± 57 | 215 ± 42 | 218 ± 22 |
| kangaroo | 1002 ± 587 | 804 ± 289 | 400 ± 278 |
| krull | 2656 ± 180 | 2333 ± 309 | 2308 ± 268 |
| kung_fu_master | 4037 ± 2962 | 9392 ± 6289 | 4031 ± 3068 |
| ms_pacman | 1053 ± 193 | 818 ± 94 | 517 ± 38 |
| pong | -20 ± 0.4 | -20 ± 0 | -21 ± 0.1 |
| private_eye | 100 ± 0 | 51 ± 59 | 1128 ± 1067 |
| qbert | 672 ± 144 | 697 ± 78 | 733 ± 291 |
| road_runner | 5426 ± 2830 | 6965 ± 6569 | 1319± 216 |
| seaquest | 387 ± 24 | 345 ± 40 | 287 ± 87 |
| up_n_down | 5123 ± 3146 | 2197 ± 231 | 2179 ± 178 |

Table 5: Standard deviations across seeds for runs presented Table 1.

| Game | KFAC+Denoised | KFAC+TRL10x | Scattering |
|---|---|---|---|
| alien | $996 \pm 180$ | $643 \pm 51$ | $441 \pm 90$ |
| amidar | $163 \pm 15$ | $98 \pm 26$ | $84 \pm 11$ |
| assault | $501 \pm 85$ | $496 \pm 129$ | $434 \pm 304$ |
| asterix | $537 \pm 96$ | $526 \pm 64$ | $502 \pm 91$ |
| bank_heist | $100 \pm 14$ | $57 \pm 36$ | $29 \pm 13$ |
| battle_zone | $8622 \pm 5358$ | $6156 \pm 1951$ | $4311 \pm 1517$ |
| boxing | $0 \pm 2$ | $-1 \pm 3$ | $-9 \pm 12$ |
| breakout | $3 \pm 1$ | $2 \pm 2$ | $2 \pm 0$ |
| chopper_command | $692 \pm 81$ | $1302 \pm 328$ | $441 \pm 80$ |
| crazy_climber | $14242 \pm 2936$ | $3546 \pm 1231$ | $740 \pm 291$ |
| demon_attack | $582 \pm 130$ | $318 \pm 168$ | $92 \pm 232$ |
| freeway | $26 \pm 0$ | $24 \pm 0$ | $19 \pm 1$ |
| frostbite | $1760 \pm 448$ | $1483 \pm 466$ | $654 \pm 709$ |
| gopher | $363 \pm 4$ | $265 \pm 67$ | $172 \pm 3$ |
| hero | $4188 \pm 1635$ | $4206 \pm 1862$ | $4127 \pm 1074$ |
| jamesbond | $263 \pm 22$ | $217 \pm 68$ | $48 \pm 10$ |
| kangaroo | $2085 \pm 2055$ | $588 \pm 5$ | $391 \pm 52$ |
| krull | $2855 \pm 156$ | $3392 \pm 2205$ | $772 \pm 560$ |
| kung_fu_master | $8481 \pm 8270$ | $7357 \pm 9200$ | $233 \pm 205$ |
| ms_pacman | $1137 \pm 180$ | $867 \pm 128$ | $613 \pm 159$ |
| pong | $-19 \pm 0.6$ | $-19 \pm 1$ | $-20 \pm 0$ |
| private_eye | $56 \pm 42$ | $100 \pm 0$ | $0 \pm 0$ |
| qbert | $731 \pm 256$ | $538 \pm 114$ | $475 \pm 161$ |
| road_runner | $4516 \pm 2869$ | $7224 \pm 4598$ | $1278 \pm 463$ |
| seaquest | $349 \pm 63$ | $520 \pm 97$ | $213 \pm 96$ |
| up_n_down | $2557 \pm 641$ | $2108 \pm 298$ | $993 \pm 244$ |

Table 6: Standard deviations across seeds for runs presented Table 2.

