# OpenReview forum: "How many weights are enough : can tensor factorization learn efficient policies ?"
_ICLR.cc/2020/Conference — Reject_

### Official Review · AnonReviewer3 · 2019-10-17
**Official Blind Review #3**

**Rating:** 1

**Review:**

This paper suggests three different disconnected ideas for improving the number of parameters of deep vision models for playing ATARI and to improve the training speed.
- Tensor-regression layer to replace fully connected layers
- Wavelet-scattering layer to replace the first convolutional layer
- Second order optimization (K-FAC)
All the ideas mentioned in this paper are existing ones (although properly attributed), so the novelty of this work is relatively low.
The paper mentions that this particular combination is "novel", but it is not clear is there is any significant synergy between these methods and why it should be considered interesting in this particular setup.
Also the paper conflates sample-efficiency with parameter-efficiency. However, there is no indication that any of these methods address sample-efficiency which would be an interesting and useful contribution.
Also the experiments are neither very conclusive nor are they easy to interpret. For example in the pong case, there is no discernable effect of the compression ratio as the highest and lowest compression give the best (and comparable) results. Also the results come without confidence intervals.

So, in general, I would consider this paper to be an uninspired combination of pre-existing ideas with weak and inconclusive experimental results: a clear reject.

**Experience Assessment:**

I have published in this field for several years.

**Review Assessment: Checking Correctness Of Derivations And Theory:**

I assessed the sensibility of the derivations and theory.

**Review Assessment: Checking Correctness Of Experiments:**

I assessed the sensibility of the experiments.

**Review Assessment: Thoroughness In Paper Reading:**

I read the paper at least twice and used my best judgement in assessing the paper.

---

> ### Author Response · Authors · 2019-11-14
> **Clarification of results and novelty**
>
> We thank Reviewer #3 for their time. We are extremely surprised that our experimental results are not clear:
>
> 1. The main results in the paper are summarized in the two bolded bottom lines of tables 1 and 2. Across 26 Atari games with 3 random seeds each, tensor factorization itself incurs little performance loss in the low-data regime up till 5 times compression, but this can be cured with second-order optimization all the way to 10 times compression, as per table 2.
> 2. Standard deviation of all these experiments results in Table 1 and 2 has been available in appendix since v1.
> 3. If the author refers to figure 3 as 'inconclusive', it's meant to provide proof of concept, even on a simple noisy algorithm (hard-max non-distributional DQN), and does show that opposite direction effects happen between extra exploration and drawdowns. This in turn motivates the experiments of section 4.2 where the net impact of the results is averaged across many Atari environments to no ambiguous conclusion.
>
> The novelty of the paper resides in its domain of application. While we would welcome prior art references, to our knowledge, applying online tensor factorization to deep RL algorithms is effectively a virgin field.
> All the more so when using data-efficient algorithms, on a complex RL domain like Atari.
>
> Consequently, we would be extremely grateful for additional constructive criticism and directions as to how to improve future iterations of the paper.

---

> > ### Comment · AnonReviewer3 · 2019-11-14
> > **Thanks for the response**
> >
> > I have went through the paper again and I still find it hard to interpret the graphs and the results. It is not clearly marked what the different algorithms are, which ones are variants of those in the paper and which are the baselines.
> >
> > However, it does not matter that much, since the main issues of the paper is lack of novelty which is impossible to fix.

---

### Official Review · AnonReviewer1 · 2019-10-23
**Official Blind Review #1**

**Rating:** 3

**Review:**

This paper investigates a list of methods to reduce the number of weights for deep RL architecture under the low-data regime. These methods include tensor regression, wavelet scattering, as well as second-order optimization (K-FAC). The experiments on the Atari games shows that by using tensor regression to replace the dense layer of the neural nets and using K-FAC for the optimization, one can reduce around 10 times of parameters without losing too much of performance.

However, I have some concerns on the novelty of this work and therefore I’m giving this paper a weak reject. Here are the reasons:

To begin with, leveraging tensor structure of the neural nets to reduce number of parameters while maintaining similar level or getting even better results have been done before, for example: Tensorizing Neural Networks (Novikov et al, 2015), Learning compact recurrent neural networks with block-term tensor decomposition (Ye et al., 2018) etc. Although the use of tensor regression might be new, the core idea is still to leverage the low rank property of the tensor and obtain a compression of the weight tensors. Moreover, why use Tucker decomposition specifically for the tensor regression? It has been proposed that using tensor train (TT) decomposition can also get very good results (see Garipov et. al. Ultimate tensorization: compressing convolutional and FC layers alike). Is it possible to investigate the use of TT decomposition for the dense layer of the deep RL architecture? Therefore the novelty for this aspect seems a bit weak for me.

The second method the authors have attempted is to swap the convolution layer of the deep RL architecture with wavelet scattering. For one particular game (demon_attack), this approach seems to outperform every other methods by a large margin. However the experiment shows that for the rest of the Atari games, there is a huge drop (45%) of performance. Therefore the significance of this approach is rather thin for me. Maybe some further investigation of the game demon_attack is needed to understand why using scattering in this game in particular gives such a huge performance boost.

Thirdly, as an approximation of the second order optimization, K-FAC does not really concern with the main theme of the paper, which is an investigation of potential weights reduction methods. It is great that the authors applied this techniques and seems to have great results. However, as the authors pointed out, K-FAC has been wildly applied in the deep RL literature, and the authors did not propose new extension for the K-FAC method, therefore the contribution of this matter is also quite thin.

Last but not least, the writing of the paper is a bit clumsy, and I was having a hard time to figure out what exactly is the proposed method. I think this paper might need some rework on the writing to describe the idea of the authors in a more clear way for the publication. Due to these reasons, I’m giving this paper a weak reject.

Some writing comments and potential writing errors (did not affect the decision):
Page 3, first line of “Tensor regression layer”, the shape of the tensor X seems to be a typo.
Also here, the definition of <X, Y>_N in the paper is to sum over the dimension of I_1…I_N, then the shape of <X, Y>_N should be K_1*…*K_x*L_1*…*L_y, without the I_N in the middle.
Also in this section, the authors mentioned Tucker decomposition for the tensor regression. However the phrasing of this sentence needs a bit rework. The usage of “For instance here”, gives the readers a feeling that Tucker is just one possible way of doing this decomposition, but not necessarily the actual decomposition for the reported experiments.
In 2.3, there is a lack of definition for  \Lambda_1 and \Lambda_2. In addition, it would be better for the general readers to add a few definitions for the terminologies in this section. For example, “circular convolution”, “wavelet filter banks” etc. I guess people with corresponding background will understand it with no problem, however I do find myself a bit lost in this section with these terminologies.
2.4 line 6, “with A and. B smaller, architecture-dependent matrices”. I think it should be “with A and B being…“
3.1, line 5, “This is all the more pressing that….”, I did not understand this sentence.
In page 6, line 3, there is a lack of definition for “compression rate”. Is it the compression rate w.r.t only the last dense layer, or w.r.t the whole network?
Figure 4 is lacking y-axis and x-axis labels.
4.2, last bullet point, “However, one must not forget that the conv layers one learns must be somehow be well adapted…”, I get what you are saying, but the sentence is a bit clumsy.
Table 1 and 2, the row name “Average” is lacking definition.

 Overall it is a good attempt to reduce the number of weights in the deep RL architecture, but I do think the novelty of this work is a bit thin and the three contributions were not tied together with the main theme of the paper. Therefore, I’m giving this work a weak reject.


**Experience Assessment:**

I have read many papers in this area.

**Review Assessment: Checking Correctness Of Derivations And Theory:**

I assessed the sensibility of the derivations and theory.

**Review Assessment: Checking Correctness Of Experiments:**

I assessed the sensibility of the experiments.

**Review Assessment: Thoroughness In Paper Reading:**

I read the paper at least twice and used my best judgement in assessing the paper.

---

> ### Author Response · Authors · 2019-11-14
> **Revision uploaded**
>
> We genuinely thank Reviewer #1 for their very thorough and helpful review.
>
> We have uploaded a revision to the paper which we hope addresses several qualms with writing and imprecisions or typos that were noted. We are also thankful for the suggestion of a signal processing refresher which we will ultimately add to the appendix, although a self-contained 'short' and intuitive exposition of wavelet filter banks is likely to be several pages long.
>
> Regarding tensor train decomposition, this is a useful method that, while more recent, we chose not to employ. It created even more complexity in the exposition of the paper in exchange for gains that did not seem significant during our preliminary testing - for instance, the reference by Novikov et al. (2015) achieves a compression factor of 7 times, whereby our work achieves between 5 and 10 times (we have clarified as per your remark that this is with respect to the total number of weights in the network). Therefore we went for the simpler method. We will consider using tensor train in the future. But the task is made difficult due to the heavy computational load on Atari.
>
> We would genuinely be grateful for examples of use of K-FAC in the RL literature besides ACKTR. We do agree that scattering results as they stand require more work.
>
> As for the novelty of the work, we would like to defend it and stress that to our knowledge, no other prior work addresses tensor factorization as an approximation method compatible with current state-of-the-art deep RL algorithms. Given the complexities of optimization in RL (see the analogy of actor-critic with GANs (Pfau and Vinyals, 2016), multilevel optimization, or saddle point formulation (Mahadevan et al. , 2014) ), and the fact the MDP exploration introduces shifts in state-space distribution, it was not a foregone conclusion that making gradient descent happen in a tiny subspace from the get-go would not hinder the process of learning to the point of non-convergence. Filling this gap in the literature and quantifying results, especially on a challenging domain like Atari, therefore appeared a necessary stepping stone to us.

---

### Official Review · AnonReviewer2 · 2019-10-23
**Official Blind Review #2**

**Rating:** 3

**Review:**

The paper aims at parsimonious reinforcement learning by employing 3
different techniques: using tensor regression layers (Kossaifi et al.,
2017b), wavelet scattering (Mallat 2011) and using K-FAC (Kingma & Ba,
2014) as the optimization method. Learning models with fewer
weights is important not only in reinforcement learning but also
in all other machine learning areas. With the combination of tensor
regression layers and K-FAC, the proposed methods give comparable
performance on several Atari games against 2 other methods, SimpPLe
and data-efficient Rainbow, while using 2 to 10 times fewer
coefficients than data-efficient Rainbow. The use of wavelet
scattering provides improvement on 1 out of 26 Atari games. The paper
also points out an interesting concentration of eigenvalues of dense
layer of a deep RL agent which provides motivation for low-rank
presentations.

While the savings in terms of coefficients is positive, the obtained
results are of little surprise. Tensor regression layers and K-FAC are
used as is without any modification while space savings and efficiency
have been reported in corresponding references. The performances of
wavelet scattering for the reported tasks are weak (better in only one
game) and the space saving is not clear. The proposed improvements
seem to be tailored to tasks with image inputs and hence reported
results are only on Atari games (possibly with sparse and low-rank
images). It is not clear if the proposed techniques can be applied to
a wider set of reinforcement learning tasks
(e.g. https://gym.openai.com/envs/#mujoco).


It would be interesting to see if we can apply the proposed methods to
other more diverse RL tasks. The performance of wavelet scattering does indeed need
more investigation and improvement. It would also be interesting to
compare the distributions of the eigenvalues of the tensor layers
versus the dense layers in deep RL which may provide insights on the
achieved savings and the compression trade-off.


I have read  the authors' rebuttals. The reviews point to a number of directions
where the contributions could be made more significant.



**Experience Assessment:**

I have read many papers in this area.

**Review Assessment: Checking Correctness Of Derivations And Theory:**

N/A

**Review Assessment: Checking Correctness Of Experiments:**

I assessed the sensibility of the experiments.

**Review Assessment: Thoroughness In Paper Reading:**

I read the paper thoroughly.

---

> ### Author Response · Authors · 2019-11-14
> **Novelty**
>
> First off, we would like to thank Reviewer #2 for their time and their review. In particular, the suggestion to compare eigenvalues of the tensor layers versus the tensor layers is very welcome, and we will implement and explore it in further versions of the paper. As more computational power gets available, we will also accordingly look to compare MuJoCo from pixels versus their standard parameterization. It is however our belief (supported by additional experiments on toy models, and the fact that scattering on pixels wasn't critical anyway) that the tensor factorization method in the dense layer will remain broadly applicable and provide similar results.
>
> The weights gains for the scattering method chosen there, deemed unclear, are presented in the paper in the legend of table 2 - to quote, 'The Scattering column also includes KFAC optimization and TRL 5x, resulting in around 10x total weights efficiency gains'.
>
> The reviewer's comment that 'While the savings in terms of coefficients is positive, the obtained results are of little surprise' has however left us perplexed. One of the motivations for writing this paper was the question of whether low-rank approximation would actually be *compatible* with the RL process. In this, and to the best of our knowledge, our paper is the first work to demonstrate proof of this concept in the RL setting. Given the complexities of optimization in reinforcement learning (see the analogy of actor-critic with GANs (Pfau and Vinyals, 2016), multilevel optimization, or saddle point formulation (Mahadevan et al. , 2014) ), and the fact the MDP exploration introduces shifts in state-space distribution, it was not a foregone conclusion than making gradient descent happen in a tiny subspace from the get-go would not hinder the process of learning to the point of non-convergence. This is one of the reasons why the eigenvalues exploration section is important and interesting, and this is all the more relevant in the small data regime we concern ourselves with ! *What is more*, even if results were not surprising ex-post, the quantification of tensor approximation error in previous deep learning works (accuracy loss) does not carry over to the RL setting, and again to our knowledge no other works have evaluated this trade-off explicitly. We therefore believe that our work opens up possibilities for light RL models that ultimately explore better.

---

### Decision · Program_Chairs · 2019-12-19

**Decision:**

Reject

**Comment:**

In this paper dense layers in deep neural networks representing policies are replaced by tensor regression layers, also by a scattering layer, and second-order optimization is considered. The paper does not have a single consistent message, and combines different techniques for unclear reason. Important related work is not cited. The presentation was found unclear by the reviewers.